# Relationship between Physical Characteristics and Morphological Features of the Articular Radius Surface: A Retrospective Single-Center Study

**DOI:** 10.3390/diagnostics14182005

**Published:** 2024-09-10

**Authors:** Reo Asai, Akira Ikumi, Yusuke Eda, Sho Kohyama, Takeshi Ogawa, Yuichi Yoshii

**Affiliations:** 1Department of Orthopedic Surgery, Tokyo Medical University Ibaraki Medical Center, Ami 300-0395, Japan; 2Department of Orthopedic Surgery, Faculty of Medicine, University of Tsukuba, Tsukuba 305-8577, Japan; 3Department of Orthopaedic Surgery, Tsukuba Medical Center Hospital, 1-3-1, Amakubo, Tsukuba 305-8576, Japan; 4Department of Orthopedic Surgery, Kikkoman General Hospital, Noda 278-0005, Japan; 5Department of Orthopaedic Surgery, National Hospital Organization Mito Medical Center, Ibaraki 311-3193, Japan

**Keywords:** preoperative planning, distal radius, volar locking plates, physical characteristics, height, weight, transverse diameter, anteroposterior diameter, three-dimensional analysis, sex differences

## Abstract

Preoperative planning is important for the osteosynthesis of distal radius fractures. Challenges arise for patients presenting with bilateral wrist injuries or a history of contralateral wrist injuries. In such cases, the estimation of the distal radius morphology and the determination of the plate size from the preoperative physical characteristics could prove beneficial. The objective of this study was to investigate the correlation between the physical characteristics and the morphology of the distal radius articular surface. A total of 79 wrist computed tomography (CT) images (41 women and 38 men) were evaluated. Physical characteristics, such as height, weight, and body mass index (BMI), were recorded. Three-dimensional CT analysis was performed to investigate the transverse and anteroposterior diameters of the distal radius. Pearson’s correlation coefficient was used to assess the relationships between height, weight, and BMI and the transverse and anteroposterior diameters of the distal radius. A moderate to strong correlation was found in the overall analysis between body height and transverse diameter (r = 0.66). There were also moderate correlations between body height and anteroposterior diameter (r = 0.45) as well as weight and transverse diameter (r = 0.41), both of which were statistically significant (*p* < 0.001). Our findings indicate a statistically significant correlation between height, weight, and morphology of the distal radius. When analyzed by sex, the correlation between body height and the transverse diameter of the distal radius was found to be relatively strong in women (r = 0.47, *p* = 0.002), suggesting that it could be a useful indicator for preoperative planning, such as estimating plate size.

## 1. Introduction

Volar locking plates (VLPs) have become the preferred choice for the osteosynthesis of distal radius fractures, a common orthopedic procedure [1]. These plates provide stable fixation, allowing for the early mobilization and optimal healing of the fractured bone. The utilization of X-ray or computed tomography (CT) scans of the unaffected wrist is often incorporated into the preoperative planning phase for osteosynthesis and corrective osteotomy [2,3]. This imaging serves as a crucial reference, providing detailed anatomical information that guides surgeons in the reduction and the accurate placement of the VLP. Strict and meticulous preoperative planning is of paramount importance to ensure that the surgery is executed safely and precisely, thereby minimizing potential complications such as malalignment, hardware failure, or tendon/nerve damage [3,4,5].

When performing preoperative planning, orthopedic surgeons often use the unaffected side as a template for the reduction position [2,6,7]. This is also commonly performed in 3D preoperative planning. However, using the unaffected wrist as an indicator of reduction presents challenges in patients with bilateral wrist injuries or a past medical history affecting the contralateral wrist [8,9]. In such cases, preoperative planning becomes difficult because the contralateral side cannot serve as an indicator of the ideal shape of the reduction, complicating the preoperative selection of the appropriate plate size. Moreover, not all facilities are equipped for CT-based 3D preoperative planning. When preoperative planning cannot be performed, implant size must be determined intraoperatively. However, the distal radius cannot be fully exposed during surgery, and the articular surface may be displaced, making intraoperative plate selection challenging in some cases. It would be advantageous if preoperative planning could be easily performed using other factors. Estimating the morphology of the distal radius and determining the plate size from the patient’s preoperative physical characteristics would reduce the operation time and minimize the risk of size mismatch between the plate and the distal radius.

Significant individual variations in the bone morphology of the distal radius articular surface have been well documented [10]. The radial inclination and palmar tilt in 3D models are both larger in women than in men [11]. The area of the distal radius articular surface is significantly larger in men than in women [11]. In addition, the width of the anterior surface in the coronal view is larger in men than in women, and the curved part of the anterior surface in men is longer and more concave than that in women [12]. The literature discusses the relationship between physical characteristics and radial morphology. Park et al. assessed the two-dimensional morphology of the distal radius on magnetic resonance imaging (MRI), revealing positive correlations between height and both transverse and anteroposterior diameters [13]. Zenke et al. investigated the distance from the extensor pollicis longus (EPL) groove to the volar cortical line of the distal radius and found a positive correlation with height [14]. Sex-based differences in the morphology of the radius have also been reported [12,15,16,17]. However, no study has investigated the association between physical characteristics and the morphology of the distal radius in men and women separately.

In this study, we hypothesized that there is a correlation between physical characteristics (such as height, weight, and body mass index [BMI]) and the morphology of the distal radius articular surface (such as transverse and anteroposterior diameters). If the size of the distal radius can be estimated from preoperative physical characteristics, it would allow for more convenient preoperative planning, reduce the necessity of additional imaging tests, and potentially shorten operation times. The aim of this study was to investigate the relationship between patients’ physical characteristics and the morphological characteristics of the distal radius joint surface using CT images of healthy wrist joints.

## 2. Methods

The study protocol was approved by an institutional review board (approval No. T2022-0041). This was a retrospective case–control study (level of evidence: III). A radiographic database was accessed to identify patients who underwent CT scans of the unaffected wrist for comparison with those of the affected side. From the database, we evaluated CT images of the unaffected wrist between January 2016 and August 2022. The absence of previous history or complaints in the unaffected wrist was confirmed through interviews and medical records. Patients with a history of traumatic arm injuries were excluded from this study. Patients younger than 18 years were also excluded from the study. A total of 79 wrist CT images, including those of 41 women and 38 men (age range: 20–95 years, mean age: 58.4 years for men; age range: 26–91 years, mean age: 62.5 years for women), were evaluated. Physical characteristics such as height, weight, and BMI were recorded for each patient at the initial hospital visit, and these data were retrieved from medical records.

### 2.1. Three-Dimensional Bone Morphology and Analysis

CT imaging and analysis of the 3D bone model of the distal radius were performed as previously described [18]. Using computer analysis software (BoneSimulater, Orthree, Osaka, Japan), we defined the long axis of the radius. The plane containing the long axis and the radial styloid process was defined as the coronal plane. The plane containing the long axis and perpendicular to the coronal plane was defined as the sagittal plane. The plane perpendicular to the long axis was defined as the axial plane. Subsequently, three reference points, (1) the radial styloid process, (2) the volar edge of the sigmoid notch, and (3) the dorsal edge of the sigmoid notch, were marked (Figure 1a). The 3D coordinates of each reference point were assessed using the 3D images. These reference points were utilized because of their high inter-rater reliability and reproducibility in our preliminary study [18]. The vertical distance between reference points (1) and (2) on the coronal plane was defined as the radius transverse diameter (Figure 1b), and the vertical distance between reference points (2) and (3) on the sagittal plane was defined as the radius anteroposterior diameter (Figure 1c). The 3D analysis was performed by an experienced hand surgeon.

### 2.2. Statistical Analysis

The results are presented as mean ± standard deviation (SD). The mean values were compared between the sexes using Welch’s *t*-test. The coefficient of variation (CV) was calculated as SD/mean. Pearson’s correlation coefficient (r value) was used to assess correlations between height, weight, BMI, and transverse and anteroposterior diameters of the distal radius. Additionally, correlation coefficients were calculated separately for men and women. Correlations were interpreted as weak (0.1 ≤ |r| ≤ 0.3), moderate (0.4 ≤ |r| ≤ 0.6), or strong (0.7 ≤ |r|) [19]. A *p*-value <0.05 was considered statistically significant. Statistical tests were performed using R software version 4.3.1 (R Foundation for Statistical Computing, Vienna, Austria).

## 3. Results

Table 1 summarizes the measurements of the physical characteristics of the patients and the morphology of the distal radius. The mean values for height and weight were larger in men than in women (*p* < 0.001). The mean value for BMI did not differ between the sexes (*p* = 0.258). In both men and women, the CVs for weight and BMI were greater than those for height. The transverse and anteroposterior diameters of the distal radius were larger in men than in women (*p* < 0.001), but there were no differences in variability.

The correlation coefficients are listed in Table 2. Scatter plots of height, weight, and BMI versus transverse and anteroposterior diameters are shown in Figure 2. In the overall analysis, a moderate to strong correlation was observed between height (x) and transverse diameter (y) (r = 0.66, y = 0.20x − 6.67), and moderate correlations were observed between height (x) and anteroposterior diameter (y) (r = 0.45, y = 0.07x + 2.05) and between weight (x) and transverse diameter (y) (r = 0.41, y = 0.08x + 21.3), all of which were statistically significant (*p* < 0.001). In the analysis by sex, a moderate correlation was found between height (x) and transverse diameter (y) in women, which was also statistically significant (r = 0.47, y = 0.14x + 2.41, *p* = 0.002).

## 4. Discussion

We investigated the relationship between physical characteristics and the transverse and anteroposterior diameters of the articular radial surface in a single institution. Statistically significant correlations were observed overall between transverse diameter and height, transverse diameter and weight, and anteroposterior diameter and height. In the analysis by sex, a moderate correlation was found only between transverse diameter and height in women.

Several studies have thoroughly investigated and reported the intricate relationship between the physical characteristics and the morphological features of the distal radius, as examined in our current study. These studies have aimed to understand how various physical attributes, such as height, influence the anatomy and dimensions of the distal radius [13,14]. For instance, sex identification using radial length has been reported to have a sensitivity of 83% and a specificity of 96% [15]. Additionally, one study indicated that the size of the radial head is useful for sex identification [16], although these studies were mostly conducted within the field of archaeology. From a more clinical perspective, reports highlight the differences in distal radius morphology between sexes [12] and suggest that it would be beneficial to create separate VLPs for men and women [1]. In our study, we observed that the patients’ overall height showed a statistically significant correlation with both the transverse and anteroposterior diameters of the distal radius, with correlation coefficients of 0.66 and 0.45, respectively. These findings suggest a meaningful association between a person’s height and the size of the distal radius. Previous studies have also obtained similar findings, indicating a connection between height and radial dimensions [13,14]; however, it is worth noting that their reference points and methodologies were different from ours, leading to variations in the results. Further investigation into the relationship between height and the morphology of the radius revealed a positive association between height and radial length [20], highlighting the potential influence of overall stature on radial dimensions. Additionally, there is a well-documented strong correlation between radial length and the transverse diameter of the distal radius, with a particularly high correlation coefficient of 0.753 reported in a study [21]. This underscores the potential interconnectedness of these anatomical features. Given these findings, it is plausible to hypothesize that height is indeed correlated with the transverse diameter of the distal radius. However, it appears that radial length may have an even stronger correlation with the transverse diameter than height alone. This suggests that while height is an important factor, radial length could serve as a more precise predictor of the transverse diameter of the distal radius. Consequently, understanding these relationships can be crucial for clinical practices, such as preoperative planning and surgical interventions, where precise anatomical knowledge is essential for optimal outcomes.

There was a moderate correlation between weight and transverse diameter (r = 0.41) and a weak to moderate correlation between weight and anteroposterior diameter (r = 0.35), both of which were statistically significant. These correlations between weight and both the transverse and anteroposterior radial diameters have not been previously reported. Weight can fluctuate throughout the day, and it is conceivable that individuals of the same height can have significantly different weights. Comparing the CV results, it is evident that weight exhibited greater variability than height. Considering that height correlates with both transverse and anteroposterior radial diameters and can influence weight, it likely acts as a confounding factor in these results. Further investigation, such as multivariate analysis, is needed to explore this relationship in more detail. The correlations between BMI and both transverse and anteroposterior diameters were weak and not statistically significant, suggesting that body shape is not a reliable indicator for predicting the morphology of the radius.

Focusing on sex differences, this study revealed interesting distinctions in the relationship between physical characteristics and distal radius morphology. It was found that only height and the transverse radial diameter in women exhibited a moderate correlation. While both men and women showed similar coefficients of variation (CVs) for height and transverse diameter, the correlation coefficient was notably higher for women. In other words, although a weak to moderate correlation between height and transverse diameter was observed in men, a stronger and more pronounced association was demonstrated in women. This suggests that in women, height may be a more reliable predictor of transverse diameter, indicating a higher accuracy in predicting the transverse diameter from height in the female population. In contrast, when examining the relationship between height and the anteroposterior diameter, the findings were somewhat different. Although a moderate correlation between height and anteroposterior diameter was observed overall in both sexes, the correlation was only weak in men and, intriguingly, no significant correlation was observed in women. This discrepancy highlights the complex interplay between height and the anteroposterior diameter, which seems to vary significantly between the sexes.

Previous studies have reported a correlation between height and anteroposterior diameter without differentiating by sex [13,14], suggesting a generalized relationship between these parameters. However, the lack of differentiation by sex in these studies may have masked important nuances. The current findings indicate that there may be differences in the growth patterns of the anteroposterior diameter between men and women, underscoring the importance of sex-specific analyses. This necessity of sex-specific analyses has been emphasized in previous research, as certain anatomical features and growth patterns may differ significantly between men and women due to genetic, hormonal, or developmental factors [1,12]. Understanding these differences is crucial for improving clinical assessments, surgical planning, and personalized treatment strategies, ultimately enhancing patient care outcomes.

In this study, the correlation coefficient between height and transverse diameter in women was 0.47, and the coefficient of determination calculated from this was 0.22. These values are still low for predicting transverse diameter from height, suggesting that while multivariate analysis combining various factors is one approach, it is also necessary to explore factors more strongly correlated with the morphology of the distal radius. The value of this study lies in the ability to predict the morphology of the radius using relatively simple anthropometric indicators such as height and weight. Although it was shown that radial length correlates more strongly with transverse diameter than height [21], this might not be useful in cases of bilateral injury. Focusing on the lower limbs, several studies in Asian populations have found a correlation between foot length and height [22,23]. Foot length can be easily measured, and feet are unlikely to be injured simultaneously in distal radius fractures, indicating that foot length may be a potentially useful indicator for estimating the shape of the distal radius in the future studies. To enhance the accuracy of estimating distal radius morphology for preoperative planning and to scale this approach to other clinical cases, several new methods could be explored. Incorporating multiple physical characteristics, such as height, weight, radial length, and other anthropometric data, into predictive models could improve preoperative planning accuracy, particularly in complex cases like bilateral wrist injuries. Advanced image analysis, incorporating machine learning algorithms, could be used to analyze larger datasets of three-dimensional CT scans to detect patterns in bone morphology. Additionally, similar approaches for correlating physical characteristics with bone morphology could be applied to other bones, such as the humerus or femur, to improve surgical planning for fractures in different areas of the body. These techniques should be considered in future studies.

It should be noted that this study has several limitations. First, this study was conducted at a single institution with a relatively small sample size of approximately 40 cases for each sex. This limits the ability to demonstrate significantly weak correlations. It would be beneficial to increase the sample size across multiple institutions in future studies. Furthermore, the correlation coefficient analysis was univariate, rendering it vulnerable to the impact of outliers. Therefore, it is necessary to develop and validate prediction models for transverse and anteroposterior diameters using multivariate regression analysis. Moreover, 3D analysis was conducted by a single examiner, and the reliability of the assessment could not be evaluated. Nevertheless, our preliminary study, which employed the same 3D analysis method, indicated minimal interexaminer error using intraclass correlation coefficients [18], suggesting that the impact on this study’s findings is likely limited.

The findings of this study indicate a correlation between height and transverse diameter of the distal radius, with a particularly moderate correlation observed in women. Further research will be conducted with a larger sample size and with the application of multivariate analyses to predict plate size based on physical characteristics.

In conclusion, this study examined the relationship between the physical characteristics of patients and the transverse and anteroposterior diameters of the radius, as observed through analysis of 3D CT images of the wrist joint. A correlation was observed between the height and the transverse and anteroposterior diameters of the distal radius. Subsequent analysis by sex revealed that this correlation was present between the height and transverse diameter in women.

## Figures and Tables

**Figure 1 diagnostics-14-02005-f001:**
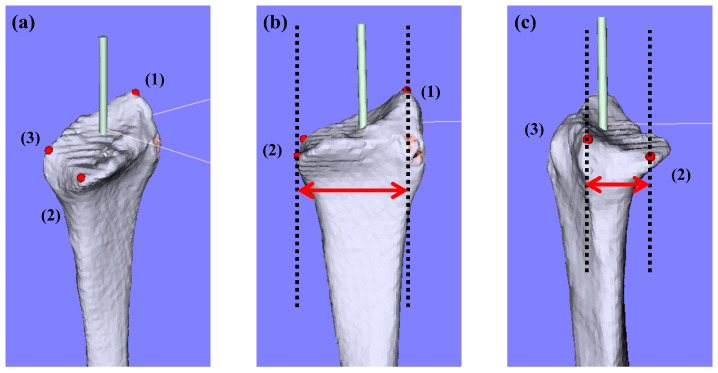
Analysis of the 3D bone model of the distal radius. Three reference points were identified: (1) the radial styloid process, (2) the volar edge of the sigmoid notch, and (3) the dorsal edge of the sigmoid notch (**a**). The radius transverse diameter was measured as the vertical distance between points (1) and (2) on the coronal plane (**b**), while the radius anteroposterior diameter was measured as the vertical distance between points (2) and (3) on the sagittal plane (**c**).

**Figure 2 diagnostics-14-02005-f002:**
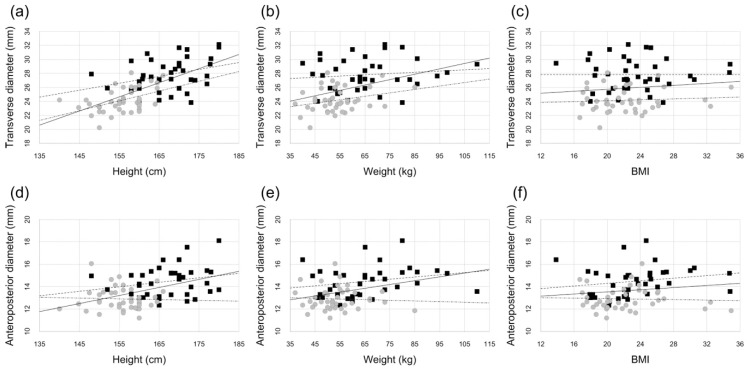
Scatter plots of physical characteristics versus distal radius morphology. Scatter plots show the relationship between transverse diameter and height (**a**), weight (**b**), and BMI (**c**), as well as between anteroposterior diameter and height (**d**), weight (**e**), and BMI (**f**). The male patients are represented by black squares, while the female patients are represented by grey circles. The regression line representing the overall correlation is depicted as a solid line, the regression line for men as a dashed line, and the regression line for women as a dot-dashed line.

**Table 1 diagnostics-14-02005-t001:** Physical characteristics and distal radius morphology of the patients.

	Overall (*n* = 79)				Men (*n* = 38)				Women (*n* = 41)			
	Mean ± SD			CV	Mean ± SD			CV	Mean ± SD			CV
Height	161.3	±	8.8	0.05	167.4	±	7.5	0.04	155.7	±	5.7	0.04
Weight	59.6	±	14.2	0.24	65.9	±	15.8	0.24	53.7	±	9.5	0.18
BMI	22.8	±	4.3	0.19	23.3	±	4.4	0.19	22.2	±	4.2	0.19
Transverse diameter	25.9	±	2.7	0.10	27.8	±	2.3	0.08	24.2	±	1.7	0.07
Anteroposterior diameter	13.7	±	1.4	0.10	14.5	±	1.3	0.09	12.9	±	1.0	0.08

**Table 2 diagnostics-14-02005-t002:** Correlation coefficients between physical characteristics and distal radius morphology.

		Height		Weight		BMI	
		r	*p*-Value	r	*p*-Value	r	*p*-Value
Transverse diameter	Overall	**0.66**	<0.001	**0.41**	<0.001	0.11	0.325
	Men	0.33	0.046	0.12	0.459	0.00	1.000
	Women	**0.47**	0.002	0.27	0.084	0.08	0.628
Anteroposterior diameter	Overall	**0.45**	<0.001	0.35	0.002	0.15	0.197
	Men	0.23	0.168	0.24	0.154	0.20	0.238
	Women	−0.04	0.808	−0.06	0.731	−0.05	0.777

Numbers in bold indicate significant correlations.

## Data Availability

The datasets analyzed during the present study are available from the corresponding author upon reasonable request.

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
