# Peer review of "Relationship between Physical Characteristics and Morphological Features of the Articular Radius Surface: A Retrospective Single-Center Study"

_diagnostics, 2024, doi:10.3390/diagnostics14182005_

Round 1

Reviewer 1 Report

Comments and Suggestions for Authors

The introduction is complete and well-written. In my opinion, it is a bit long and could be shortened as advised:

Lines 53-65 are useless in the introduction of this study

Lines 91-94 would be more appropriate in discussion (leaving a sentence like in “our knowledge no study has investigated…”)

Materials and Methods are complete and clear

Results are clear with good tables and graphisms

Discussion is ggod.

Author Response

The introduction is complete and well-written. In my opinion, it is a bit long and could be shortened as advised:

Lines 53-65 are useless in the introduction of this study

Response) It was deleted.

Lines 91-94 would be more appropriate in discussion (leaving a sentence like in “our knowledge no study has investigated…”)

Response) It was moved to the discussion section. (Page 6, Line 175-180)

Materials and Methods are complete and clear

Results are clear with good tables and graphisms

Discussion is ggod.

Reviewer 2 Report

Comments and Suggestions for Authors

Dear Authors,

This retrospective study of yous seems quite interesting. However, as I mentioned before, each statement written in a scientific paper should be justified by at least a relevant reference. Please add references on lines 43-45, lines 47-50, lines 57-59, lines 59-61, lines 66-68, lines 68-70, lines 73-76.

Line 17: Using the term "However" is expressed some kind of opposition. So there is no need for this term in line 17.

Lines 27-28: Please define which height, person or distal radius?

Lines 52-53: Please mention the name, version and manufracturer of the software used.

Lines 81-82: Please mention in short these variations.

Line 181: Discussion (without s) is more appropriate.

Author Response

This retrospective study of yous seems quite interesting. However, as I mentioned before, each statement written in a scientific paper should be justified by at least a relevant reference. Please add references on lines 43-45, lines 47-50, lines 57-59, lines 59-61, lines 66-68, lines 68-70, lines 73-76.

Response) The references were added. Since one reviewer suggested to delete for the sentences of Line 53-65, it was deleted.

Line 17: Using the term "However" is expressed some kind of opposition. So there is no need for this term in line 17.

Response) It was deleted.

Lines 27-28: Please define which height, person or distal radius?

Response) In our study, "height" refers to the patient's body height, not the height of the distal radius. Now it was clarified in the text. (Page 1, Line 28-33)

 Lines 52-53: Please mention the name, version and manufracturer of the software used.

Response) Since one of the reviewers suggests to delete this paragraph, this description was deleted.

Lines 81-82: Please mention in short these variations.

Response) Descriptions for the variation were added in the text. (Page 2, Line 67-72)

Line 181: Discussion (without s) is more appropriate.

Response) The ‘s’ was deleted.

Reviewer 3 Report

Comments and Suggestions for Authors

If we consider that this article is only the beginning of the study of the global problem the optimal choice of implant size for osteosynthesis or joints arthroplasty, then, in general, it can be recommended for publication in the presented form.

At the same time, it should be noted that the amount of data selected for statistical analysis of the problem is insufficient, even from the point of view of this study, which is recognized, among other things, by the authors themselves.

To improve the quality of the article, it is desirable to briefly describe the new approaches that should be taken for a more accurate solution of the problem in the future and the possibility of scaling it to other clinical cases.

Author Response

If we consider that this article is only the beginning of the study of the global problem the optimal choice of implant size for osteosynthesis or joints arthroplasty, then, in general, it can be recommended for publication in the presented form.

At the same time, it should be noted that the amount of data selected for statistical analysis of the problem is insufficient, even from the point of view of this study, which is recognized, among other things, by the authors themselves.

To improve the quality of the article, it is desirable to briefly describe the new approaches that should be taken for a more accurate solution of the problem in the future and the possibility of scaling it to other clinical cases.

Response) Thank you for the constructive comments. The descriptions for the future study were added in the text. (Page 7, Line 254-264)